# Changes in Whole Blood Polyamine Levels and Their Background in Age-Related Diseases and Healthy Longevity

**DOI:** 10.3390/biomedicines11102827

**Published:** 2023-10-18

**Authors:** Kuniyasu Soda

**Affiliations:** 1Saitama Medical Center, Jichi Medical University, Saitama 330-8503, Japan; soda@jichi.ac.jp; 2Saitama Ken-o Hospital, Saitama 363-0008, Japan

**Keywords:** polyamine, spermine, spermidine, lifespan extension, polyamine-rich food, age-related disease, inflammation, gene methylation, LFA-1

## Abstract

The relationship between polyamines and healthy longevity has received much attention in recent years. However, conducting research without understanding the properties of polyamines can lead to unexpected pitfalls. The most fundamental consideration in conducting polyamine studies is that bovine serum used for cell culture contains bovine serum amine oxidase. Bovine serum amine oxidase, which is not inactivated by heat treatment, breaks down spermine and spermidine to produce the highly toxic aldehyde acrolein, which causes cell damage and activates autophagy. However, no such enzyme activity has been found in humans. Polyamine catabolism does not produce toxic aldehydes under normal conditions, but inflammation and some pathogens provoke an inducible enzyme, spermine oxidase, which only breaks down spermine to produce acrolein, resulting in cytotoxicity and the activation of autophagy. Therefore, spermine oxidase activation reduces spermine concentration and the ratio of spermine to spermidine, a feature recently reported in patients with age-related diseases. Spermine, which is increased by a long-term, continuous high polyamine diet, suppresses aberrant gene methylation and the pro-inflammatory status that progress with age and are strongly associated with the development of several age-related diseases and senescence. Changes in spermine concentration and the spermine/spermidine ratio should be considered as indicators of human health status.

## 1. Introduction

The history of the discovery and study of the existence of polyamines is extremely long. There are many findings that have accumulated during this long period of research, especially regarding the properties of polyamines, which have been clarified by many researchers, especially in the last half century or so. In the past, the relationship between cancer and polyamines was the focus of much attention and research. And our group was the first in the world to report the health and longevity effects of a diet high in polyamines, having discovered the physiological activities that may contribute to healthy longevity through basic research based on the established properties of polyamines [1]. Since that time, attention has begun to focus on the relationship between healthy longevity and age-related conditions that affect longevity. However, given the current situation, it seems that many researchers do not fully understand these basic characteristics and therefore seem to be repeating the same thing that was done before the basic background of polyamines was elucidated more than half a century ago. For example, following our report, polyamines have been reported to prolong lifespan [2,3,4,5], but some of the changes caused by the activity of polyamine degradation products have been mistaken for the biological activity of the polyamines themselves.

In order to not waste the research results that polyamine researchers have accumulated over the past half century, and not to waste more effort, time, and resources, I will devote much of the first half of this review to a detailed description of the properties of polyamines. This review is not a commentary on polyamine metabolism in general, but summarizes the basic properties of polyamines in relation to polyamine concentrations and their relationship to healthy longevity and age-related diseases, and details the properties that require attention in polyamine research and issues related to experimental methods. Then, I will review recent reports on the relationship between human health and age- and lifestyle-related diseases and polyamines, especially their concentrations, and explain the mechanisms of polyamine concentration changes based on the basic background of polyamines.

## 2. Polyamines

Natural polyamines are low-molecular-weight aliphatic polycations found in the cells of all living organisms and their intracellular concentrations are very high, ranging from micromolar to low millimolar. They are known to be essential for cell growth and differentiation and are synthesized in cells on demand. In addition to de novo intracellular synthesis, cells can take up polyamines from the environment. Polyamines (spermine and spermidine) and their precursor, putrescine, contain multiple amino groups (-NH_2_). The molecular weights of spermine (SPM) with four amino groups and spermidine (SPD) with three amino groups are about 140 g/mol and 200 g/mol, respectively. Putrescine (PUT) with two amino groups is a precursor of polyamines and is called a diamine. SPD and SPM are then synthesized by the sequential addition of aminopropyl groups donated by decarboxylated S-adenosylmethionine (dcSAM), which is converted from *S*-adenosylmethionine (SAM) by the enzymatic activities of adenosylmethionine decarboxylase (AdoMetDC). The addition of aminopropyl groups to PUT and SPD is catalyzed by spermidine synthase and spermine synthase, respectively (Figure 1).

Polyamines and putrescine are universally required for cell growth and differentiation. However, their importance has been found to vary from organism to organism, as summarized in my previous review [6]. For example, PUT is essential for cell growth in lower organisms such as bacteria and fungi, whereas spermine is not present in the cell. Or, in yeast and nematodes, spermine can be found in small amounts, but it is not essential for growth. All of this suggests that these are inconsequential roles for spermine in cell growth and differentiation, as well as cell function, in lower primitive organisms. Spermine is considered to be more important in highly developed animals. For example, a decrease in the spermine levels due to a deficiency in spermine synthase (Snyder–Robinson syndrome) has serious consequences in humans [7].

The degradation pathway of polyamines is also elucidated (Figure 2). Spermidine/spermine N-(1)-acetyltransferase (SSAT) and N1-acetylpolyamine oxidase (APAO) are enzymes that break down SPM and SPD in cells. SSAT is a highly inducible enzyme that catalyzes the transfer of acetyl groups from acetyl-coenzyme A to the terminal amines of SPM and SPD. APAO catalyzes the oxidation of N1-acetylspermine and N1-acetylspermidine produced by SSAT activity, releasing aldehyde and hydrogen peroxide to produce SPD and PUT, respectively. The resulting aldehyde, 3-acetamidopropanal, has been shown to have no cytotoxic activity [8]. This degradation pathway, along with a mechanism for transporting polyamines across the cell membrane, keeps the intracellular concentration of polyamines constant.

The alternative polyamine degrading enzyme, spermine oxidase (SMO), can directly convert SPM back to SPD (Figure 3). SMO, a highly inducible enzyme, specifically oxidizes spermine. The enzymatic activity of SMO degrades spermine, producing 3-aminopropanal (3-AP) as a by-product. Produced 3-AP spontaneously deaminates to form acrolein [9]. Unlike the metabolite (3-acetamidopropanal) produced by the enzymatic activities of SSAT and APAO, both 3-AP and acrolein are substances with potent cytotoxic activities. In fact, SMO activation in the presence of SPM has been shown to cause severe damage to cells, supporting the strong cytotoxic activity of these two substances produced by SMO [10,11]. And several pathologies in which SMO is activated have been reported [12,13,14,15], and acrolein is detected in the blood in such pathologies [16], indicating that the enzymatic activities of SMO are activated in such conditions. Conversely, the suppression of SMO activity can ameliorate disorders caused by pathological conditions [17].

## 3. The Most Basic and Important Aspects of Conducting Polyamine Research

Recently, several researchers have postulated that spermidine activates autophagy function and that its bioactivity is the mechanism through which increased polyamine intake promotes longevity [2,3,4,5]. However, these studies used medium supplemented with fetal bovine serum (FBS) to perform polyamine experiments. Importantly, ruminant serum contains a copper-containing amine oxidase called bovine serum amine oxidase (BSAO). This enzymatic activity is not observed in humans or non-ruminants.

It has been known for about 70 years that polyamines are cytotoxic when added to cell cultures mixed with FBS [18]. Cytotoxic substances are produced during the conversion of spermine and spermidine catalyzed by copper-containing amine oxidase in FBS, i.e., BSAO. This enzymatic activity does not disappear even in serum deactivated by heat treatment. BSAO catalyzes the oxidation of SPM and SPD to produce SPD and PUT, respectively, while simultaneously producing acrolein as a by-product (Figure 4). As described above, the cytotoxic activities of acrolein are very potent. Therefore, when cells are cultured with polyamines, even at low concentrations, in culture medium supplemented with FBS, acrolein provokes strong cytotoxic activity and kills cultured cells, or at least impairs cellular function.

This means that SPD is degraded to produce acrolein in the culture medium, but the production of acrolein via SPD degradation has not been confirmed in vivo. In addition, while SMO breaks down one molecule of SPM to produce one molecule of acrolein, BSAO converts one molecule of SPM to one molecule of acrolein and one molecule of SPD, which in turn breaks down the SPD produced to produce one molecule of acrolein. Simply put, SPM should be at least twice as cytotoxic as SPD in FBS-supplemented medium. And it has been reported that treatment with SPM in culture supernatant supplemented with BSAO is more toxic than SPD treatment [19]. Therefore, when cultured with SPD, the autophagic function is activated by relatively mild cytotoxic activity and the cells do not die. However, it is easy to speculate that the addition of SPM at the same concentration as SPD will kill most of the cultured cells due to increased cytotoxic activity, thus eliminating the means to determine the activation of intracellular autophagy. I suspect that this is the reason why autophagy activation has been reported by SPD, which has lower biological activities in humans, but not by SPM, which has strong biological activities and plays important roles in humans.

At this time, when the existence of enzymes that degrade SPD to produce acrolein is unknown, it can be said that at least an examination of the biological activity of SPD in culture medium mixed with FBS containing BSAO is not an examination of the biological activity that occurs in vivo in humans. Applying the reaction that occurs under these culture conditions to polyamine activity in vivo, the phenomenon is attributed to SPM degradation due to SMO activation induced by certain pathological conditions or the presence of inflammation (Figure 5). The finding of significant age-related increases in autophagy markers in the aged kidney suggests the compensatory activation of basal autophagy in response to the increased cytotoxic activities with aging [20] and supports an increase in various stimuli, such as chronic inflammation, with aging and the activation of autophagy responses induced by them. Spermine and spermidine themselves do not activate autophagy, indicating that the increase in autophagy markers upon the addition of SPD to cultured cells is only observed in the presence of BSAO [19]. Scientists need to know what is described above in order to study the activity of polyamines in vitro. We have constantly used SAO-free proteins such as human serum in our experiments. Alternatively, when using a culture medium mixed with bovine serum, researchers should mix a compound that inhibits SAO activities and study the function of polyamines [19]. In this case, however, it is important to remember that the activity of the compound itself must also be considered.

Polyamines are involved in various cellular functions such as transcription, RNA modification, protein synthesis, and the regulation of enzyme activity, and exist in association with DNA, RNA, and various protein molecules. As summarized in my previous review article [6], a high percentage of all polyamines are bound by ionic interactions to nucleic acids, proteins, and other negatively charged molecules in the cell. Therefore, very few polyamines are free in the body. When measuring polyamine concentrations in serum or plasma using HPLC, polyamine peaks, especially SPM peaks, can be difficult to detect due to the very low levels [21]. If the concentration of polyamines, especially SPM, in serum or plasma is not high, HPLC may only detect a shimmer of the baseline or no peaks at all. Therefore, determining polyamine concentrations from an uncertain peak is difficult to measure accurately, especially with respect to SPM concentrations. Many recent papers have often measured serum or plasma concentrations and have not mentioned or described well-considered SPM concentrations, suggesting that this is due to the above problems preventing adequate studies. In addition, it is important to remember that any amount of hemolysis will release large amounts of polyamines present in the blood cells into the fluid component of the blood, which will significantly distort the measurements.

We have measured polyamine levels in whole blood for the above reasons. Measuring polyamines in whole blood requires instrumental adjustments and a somewhat complicated procedure, but we believe it is essential for accurate concentration measurements.

## 4. Age-Related Changes in Polyamine Concentrations

The relationship between aging and polyamine levels has already been summarized in my previous review article [6]. During fetal and developmental stages, polyamine synthase is highly activated, but its activity gradually declines with age. From this, it can be inferred that polyamine levels decrease with age. In fact, when the relationship between age and polyamine levels is examined for all age groups, including the developing years, blood polyamine levels decrease with age, mainly as noted in the titles and abstracts of the papers [22]. However, when measured in healthy adults who have stopped growing, polyamine levels in tissue, blood, and urine have not been found to decrease with age [6,23,24]. Intracellular polyamine concentrations appear to be tightly controlled by synthesis and degradation, as well as uptake and release across the cell membrane. The absence of a decrease in intracellular polyamine concentration with aging suggests that the polyamine homeostasis mechanism plays an important role.

Although there is no decrease in polyamine levels in the body with age, it is noteworthy that there are large individual differences in blood polyamine levels [23,24]. It is not clear what the biological basis for the large individual differences in blood polyamine levels is. However, this large individual variation in polyamine concentrations is an aspect that makes the clinical application of polyamines difficult. It is well known that in cancer patients, polyamines, which are synthesized in large amounts by cancer cells, are transferred into the blood, resulting in elevated blood polyamine levels. Therefore, attempts have been made to diagnose the presence of cancer based on differences in polyamine levels, but large individual differences have made clinical application difficult. When considering the clinical application of polyamine concentration as an indicator, it is essential to study it in many clinical cases. Otherwise, large individual differences in polyamine concentrations would result in very different analysis results depending on case selection. For example, one group of investigators reported that cognitive decline was correlated with low SPD serum levels [22], while another group reported that the more severe the cognitive decline, the higher the serum SPD [25].

## 5. Age- and Disease-Related Changes in the Ratio of Spermine to Spermidine

In adults, changes in tissue and blood polyamine concentrations with age are not pronounced and do not decrease with age. However, SPM concentrations show a slight tendency to gradually decrease with age, so that the SPM/SPD ratio tends to decrease [23,26,27]. And this decline tends to be more pronounced in patients with age- and lifestyle-related diseases [23,26]. In particular, it has been reported to be more pronounced in patients with renal failure. SPD concentrations in erythrocytes were found to be significantly higher in patients with advanced renal failure who were not on hemodialysis compared to healthy subjects. In these patients, erythrocyte SPM concentrations were unchanged as compared to healthy subjects, resulting in a lower SPM/SPD ratio [28].

Similar changes in polyamine levels, i.e., a decrease in the SPM/SPD ratio and/or an increase in SPD levels, have been reported in other age-related diseases such as cerebral infarction, neurodegenerative diseases, and sarcopenia [26,27,29,30]. In patients with neurodegenerative diseases such as Alzheimer’s disease, SPD levels were elevated in the frontal and parietal lobes of the brain [31]. Plasma concentrations of PUT and SPD increased in stroke patients, while SPM concentrations remained unchanged, resulting in a significant decrease in the SPM/SPD ratio [32]. Similarly, it has been reported that the blood SPM/SPD ratio of Parkinson’s disease patients is lower than that of healthy subjects, and that the age-related decline in the SPM/SPD ratio occurs at a younger age and is more pronounced than in healthy subjects [26]. We also found that whole blood SPD levels were higher and SPM/SPD ratios were lower in sarcopenic patients than in non-sarcopenic subjects. And the SPM/SPD ratio tended to decrease with age in sarcopenic patients, while no such decrease was observed in the non-sarcopenic elderly population [23].

Chronic inflammation has been implicated in the background of these age-related chronic diseases [33]. This means that in patients with these diseases, inflammation-induced increases in SMO activity and 3-AP or acrolein levels should be noted. In fact, urinary acrolein levels were significantly higher in patients with diabetes mellitus than in those without [34]. An increase in plasma acrolein concentration and SMO activity has been observed in patients with chronic renal failure, such as diabetic nephropathy, chronic glomerulonephritis, and nephrosclerosis [35]. In patients with cerebrovascular disease, SMO was activated [17,32] and plasma acrolein levels were elevated [36,37]. Plasma acrolein levels were higher in patients with rheumatoid arthritis than in healthy individuals [38]. At the same time, in patients with brain damage, increases in plasma acrolein concentrations were associated with increases in interleukin-6 and C-reactive protein [37], suggesting a close relationship between inflammation, SMO activation, and acrolein production.

Chronic inflammation has been implicated in the onset and progression of several age- and lifestyle-related diseases, as well as protein-energy depletion leading to cardiovascular disease and sarcopenia [39,40,41]. In addition, the presence of chronic inflammation was a strong predictor of poor outcomes in dialysis patients [42]. In light of these scientific facts, the cytotoxic activity of acrolein, which results from the degradation of SPM via activated SMO in the presence of inflammation, may contribute to the development and progression of these diseases and the prognosis of patients [43,44].

As noted above, in diseases with a background of chronic inflammation, SPM is degraded, resulting in a lower SPM/SPD ratio. Therefore, the SPM/SPD ratio may have clinical application as a predictor of the onset and progression of age- and lifestyle-related diseases. At the same time, however, the SPM/SPD ratio in the blood, as well as the concentrations of SPM and SPD, vary widely among individuals, and it may be difficult to determine the risk of disease development and progression using a single blood test. However, it may be possible to assess the risk of disease onset and determine disease severity by tracking the SPM/SPD ratio over time in the same individual.

## 6. Polyamines as Nutritional Contributors to the Prevention of Age- and Lifestyle-Related Disease Development

Considering that the main source of polyamines is thought to be the gastrointestinal tract, i.e., polyamines in food and polyamines synthesized by intestinal bacteria are important, and that cells can take up extracellular polyamines, it is likely that polyamine levels in the body are affected by food intake and the state of intestinal bacteria. In fact, many studies have shown that reducing polyamine intake, as well as inhibiting the activity of gut bacteria with antibiotics, reduces blood polyamine levels [45,46]. Conversely, there has been little experimentation on how polyamine delivery to the gastrointestinal tract affects polyamine levels in the body, but long-term consumption of a diet high in polyamines will gradually increase blood polyamine levels [47,48]. However, it has also been found that a diet high in polyamines does not affect body concentrations in the short term [24,48,49].

When SPD was mixed with drinking water and administered to mice, an increase in blood SPD levels was reported [2]. SPM concentrations also appeared to have increased, as seen in the figure in the paper, but specific data are not shown. Interestingly, our first preliminary experiments with a small number of mice also showed a significant increase in SPD. However, this is not a study that followed changes in concentration in individual animals, but rather a population study with a small number of animals, so it is difficult to deny that the study captured differences due to the chance selection of cases. In fact, the standard deviation of the mean blood concentration was extremely large, and the individual differences in concentration appeared to be extremely large too.

We found that when mice were fed a diet containing synthetic polyamines and a polyamine concentration about three times higher than that of soybeans for a long period of time, the blood SPM concentration gradually increased, with a significant difference in the 25th week of feeding [1]. Blood SPD levels also increased in some animals, resulting in a slight increase in mean SPD, but individual differences in SPD concentrations increased and were not significant. The effects of large differences in polyamine concentrations have a common background in that the choice of cases to study in relation to the disease can yield quite opposite results [22,25].

There are not many human intervention studies using high polyamine diets. The results of studies using high SPD supplements have been reported [49]. There was a 12-month study of supplementation in elderly patients between the ages of 60 and 90. This study showed a 10–20% increase in polyamine intake compared to normal dietary polyamine intake, yet the blood SPD levels did not change at all [49]. Thus, if there are clinical changes after being on a high SPD diet, it is not at all clear whether they are due to SPD or to other components that occurred at the same time [50]. Specific and detailed data on changes in SPM concentrations are not clear in the report, but the figure in the paper shows a slightly increasing trend in SPM concentrations.

A study of changes in polyamine concentrations in plasma and saliva due to high doses of SPD supplements has recently been reported. The study was conducted as a randomized, placebo-controlled, crossover trial and demonstrated that SPD supplementation increased plasma SPM levels. However, plasma SPD and PUT levels were unchanged [51].

We conducted a study of long-term, high-concentration polyamine diets in humans [24]. The results showed that SPM levels gradually increased, with a significant difference in SPM levels after one year. In this study, natto (fermented soybeans), a traditional Japanese food, was used and the subjects were given almost the same amount of polyamines as found in their regular diet. In other words, the amount of polyamine intake was almost double that of the regular diet. And our reported bioactivity of spermine, i.e., the suppression of lymphocyte function-associated antigen 1 (LFA-1) expression on immune cells, was confirmed in association with changes in SPM concentration [24]. Namely, changes in each individual’s blood SPM concentration were negatively correlated with changes in LFA-1 expression.

Because polyamines are absorbed from the gastrointestinal tract without being broken down, and because many foods generally contain more SPD than SPM, it was thought that a diet high in polyamines would increase blood SPD. However, although limited research can be confirmed, there is very little evidence that continuous consumption of a high polyamine diet increases the blood levels of SPD. Instead, prolonged consumption of a diet high in polyamines (richer in SPD than in SPM) or even a short period of high SPD intake appears to increase SPM concentration, although there are individual differences. It is interesting to note that the age- and lifestyle-related diseases, which are associated with shorter life expectancy, decrease the SPM/SPD ratio, whereas diets rich in polyamines, which contribute to life extension, such as soy products, increase the SPM/SPD ratio.

## 7. Biological Activity of Polyamines in Human Health and Disease

Polyamines are known to possess many biological activities that may counteract age-related conditions and senescence [6]. For example, they have anti-inflammatory and antioxidant properties [52,53] and protect cells and genes from damaging stimuli such as ionizing radiation, ultraviolet rays, toxic chemicals, and other stresses [6]. And some researchers, including us, have reported that increasing polyamine intake extends the lifespan of animals [1,5,54].

The anti-inflammatory effects of polyamines include the suppression of proinflammatory cytokine production by immune cells upon the stimulation and suppression of LFA-1 expression in the cell membrane [52,53]. Increased LFA-1 protein causes immune cells to respond to even minor stimuli, triggering the production of proinflammatory cytokines and provoking inflammation. SPM has strong physiological activity and therefore shows anti-inflammatory activity over a range of physiological concentration changes. SPD also shows similar biological activity to SPM, but requires a concentration change well beyond the physiological concentration change to confirm the effect [52,53]. Furthermore, the suppression of LFA-1 expression on immune cells by SPM is specific [52].

The amount of LFA-1 has been found to be related to the methylation status of the ITGAL, where the gene for LFA-1 is encoded. Increased levels of LFA-1 protein on immune cells with aging are associated with the progressive demethylation of ITGAL [55,56]. And in our in vitro experiments, the demethylation of ITGAL was associated with an increase in the amount of LFA-1 protein on immune cell membranes, and conversely, the hypermethylation of ITGAL was associated with a decrease in LFA-1 protein levels [57].

Gene methylation is a change that only occurs in cytosine, one of the four bases that make up a gene’s information, and is a mechanism that alters the reading of genetic information by adding or removing methyl groups from cytosine. In front of the genetic information, there is a site called the CpG land, which contains repeated sequences of cytosine and guanine. The methylation of cytosines within the CpG island results in decreased transcription and consequently decreased production of the protein encoded by the gene. Conversely, when cytosines within the CpG island are demethylated, transcription is more likely to occur, resulting in increased synthesis of the protein encoded by the gene. DNA methylation is regulated by DNA methyltransferases (DNMTs). DNMTs control the methylation state of cytosines by using methyl groups provided by SAM (Figure 6). The methyl group donor, SAM, on the other hand, is converted to decarboxylated S-adenosylmethionine (dcSAM) by S-adenosylmethionine decarboxylase (AdoMetDC). In the synthesis of SPD and SPM, aminopropyl groups are provided by dcSAM. dcSAM is a potent inhibitor of DNMT, and as dcSAM increases, DNMT activity decreases [58,59]. It has also been reported that there is an inverse correlation between the dcSAM/SAM ratio and DNMT activity [60]. Reduced DNMT activity not only results in demethylation due to a decreased ability to donate a methyl group to cytosine, but also in progressive demethylation at one site and hypermethylation at another, resulting in aberrant methylation of the entire genome [57,61,62]. We found that a decrease in DNMT activity increased the demethylation of the ITGAL region, but at the same time created a genome-wide condition called aberrant methylation, in which demethylation and methylation occur at different sites in the gene. Conversely, when DNMT is activated, the entire genome is no longer aberrantly methylated and ITGAL becomes highly methylated [57] (Figure 6).

The decrease in the activity of ODC due to aging and the decrease in the concentration of SPM due to degradation by chronic inflammation cause the activation of AdoMetDC, resulting in the production of more dcSAM and the decrease in the activity of DNMT, which is repressed by dcSAM, leading to aberrant methylation of the entire gene and demethylation of ITGAL. The degradation of SPM by SMO leads to the formation of acrolein, which has potent cytotoxic activity, and at the same time, coupled with increased LFA-1 protein levels due to the demethylation of ITGAL in association with aberrant methylation of the entire genome, induces a proinflammatory state and induces inflammation and cellular damage. Conversely, SPM, which is increased by a diet high in polyamines, suppresses the activity of AdoMetDC and increases the supply of methyl groups to genes due to its strong physiological activity (negative feedback mechanism). As a result, DNMT is activated and the methylation status of the entire genome is regulated. At the same time, ITGAL is hypermethylated, and LFA-1 protein levels decrease. That is, the increased polyamine intake acts to inhibit aberrant methylation of the entire genome, protecting cells and genes from harmful stimuli, and suppressing the proinflammatory state. The opposite occurs in diseases caused by chronic inflammation.

## 8. Conclusions of the Review and Issues to Be Addressed

In the previous half of this review, I provided basic notes on polyamine research that researchers need to understand. In addition, I outlined the possibility that changes in blood polyamine concentrations, particularly the SPM/SPD ratio, may be an indicator of the onset and progression of age- and lifestyle-related diseases, as well as the physiological activity of polyamines that may underlie the effect of increased SPM concentration on lifespan extension due to continuous consumption of a high polyamine diet (Figure 7).

There are several things that need to be resolved in this area. One is the cause of individual differences in polyamine concentrations. Intracellular polyamine concentrations are strongly influenced by extracellular supply. The largest source of this supply is the gastrointestinal tract. Therefore, it is necessary to investigate which elements of the gastrointestinal tract cause differences in polyamine supply. Second, it is primarily spermine that is increased by a polyamine-rich diet, e.g., natto, a fermented soybean food, even though the diet normally contains high levels of spermidine. It is necessary to investigate which factors in the intestinal environment promote spermine synthesis as a possible cause of individual differences in polyamine concentrations. It would be necessary to investigate which factors in the intestinal environment increase spermine supply, as well as the causes of individual differences in polyamine concentrations. It is interesting to note that most intestinal bacteria can only synthesize up to spermidine. We consider the regulation of gene methylation and the associated anti-inflammatory effects to be the background of polyamine-induced healthy longevity, but what other mechanisms might contribute?

## Figures and Tables

**Figure 1 biomedicines-11-02827-f001:**
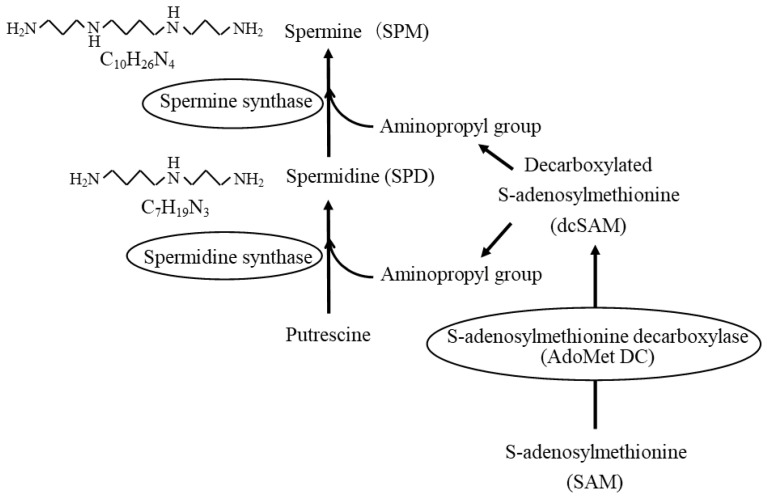
Polyamine synthesis pathway. Spermidine synthase and spermine synthase catalyze the addition of aminopropyl groups to putrescine and spermidine to synthesize spermidine and spermine, respectively. Aminopropyl groups are donated by decarboxylated S-adenosylmethionine (dcSAM), which is converted from *S*-adenosylmethionine (SAM) by the enzymatic activities of adenosylmethionine decarboxylase (AdoMetDC).

**Figure 2 biomedicines-11-02827-f002:**
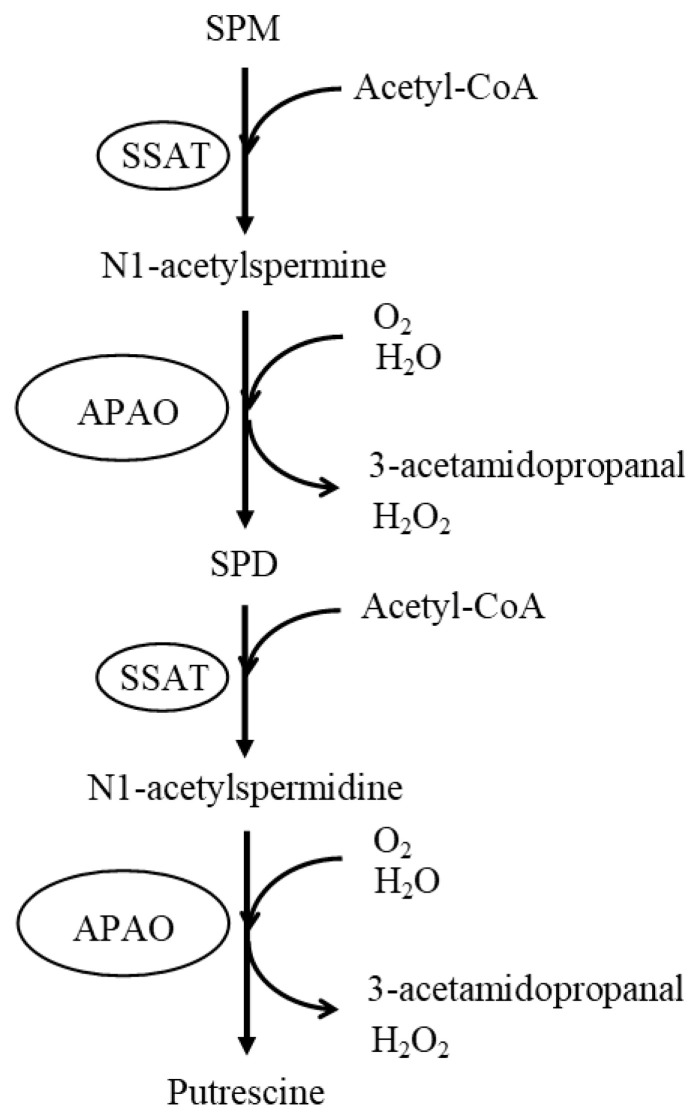
Polyamine degradation pathway under normal conditions. SPM and SPD are converted to N1-acetylspermine and N1-acetylspermidine, respectively, via the enzymatic activity of SSAT. N1-acetylpolyamine oxidase (APAO) preferentially catalyzes the oxidation of N1-acetylspermine and N1-acetylspermidine to SPD and putrescine, respectively. This degradation process produces non-toxic 3-acetamidopropanal. Abbreviations: SPM, spermine; SPD, spermidine; SSAT, spermidine/spermine N-(1)-acetyltransferase; APAO, N1-acetylpolyamine oxidase.

**Figure 3 biomedicines-11-02827-f003:**
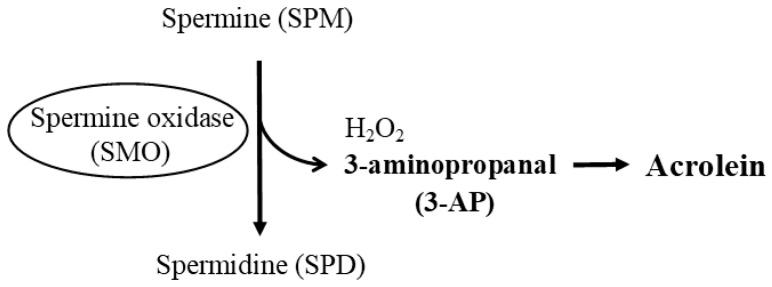
Polyamine degradation via spermine oxidase (SMO). SMO, a highly inducible enzyme expressed in macrophages and epithelial cells, directly converts SPM back to SPD. SMO degrades SPM and produces 3-AP as a by-product. Generated 3-AP is spontaneously deaminated to form acrolein. Both 3-AP and acrolein are highly toxic and exhibit cytotoxic activity. Abbreviations: SPM, spermine; SPD, spermidine; SMO, spermine oxidase; 3-AP, 3-aminopropanal.

**Figure 4 biomedicines-11-02827-f004:**
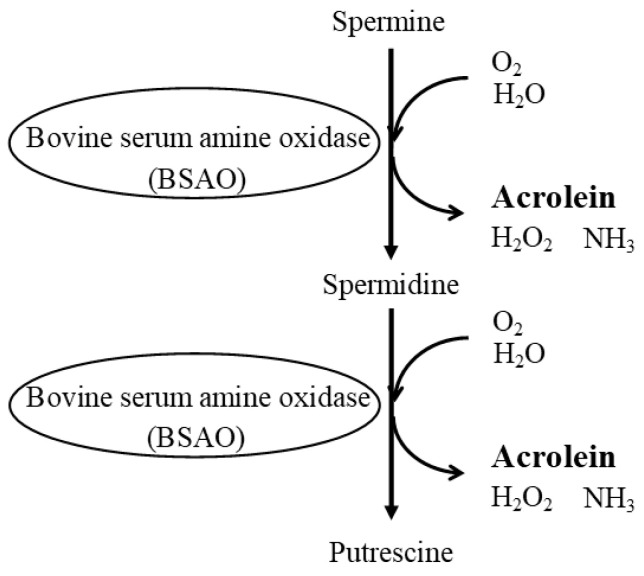
Polyamine degradation pathway via fetal bovine serum amine oxidase. BSAO in ruminant sera such as bovine serum catalyzes the oxidation of spermine and spermidine to spermidine and putrescine, respectively. The acrolein produced in this process is very toxic and can cause cell damage and death. Abbreviation: BSAO, bovine serum amine oxidase.

**Figure 5 biomedicines-11-02827-f005:**
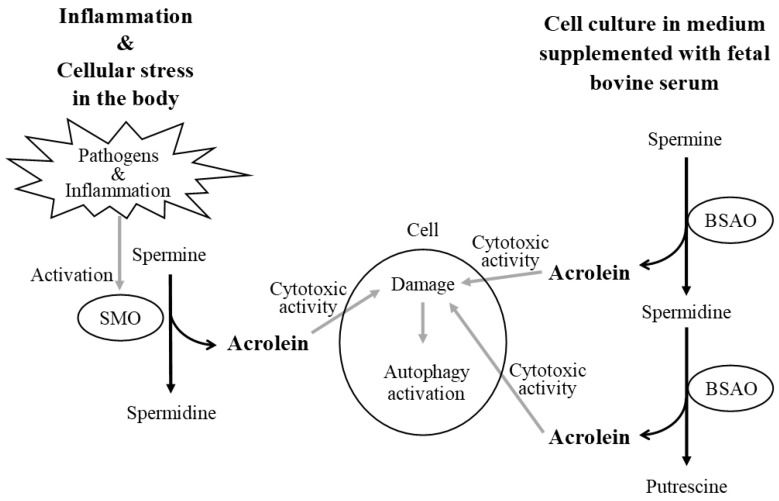
Similarities between cells cultured in bovine serum medium in vitro (**right**) and cells under cytotoxic stress such as inflammation in vivo (**left**). (**left**): BSAO in bovine serum degrades both spermine and spermidine in the culture medium to produce acrolein. Heat treatment to deactivate the serum does not eliminate the enzymatic activity of BSAO. In response to the potent cytotoxic activity of acrolein, cells have been shown to activate autophagy. Spermine and spermidine themselves have no cytotoxic activity and therefore do not activate autophagy. (**right**): Inflammation and certain pathogens activate SMO, and SMO exclusively breaks down spermine and produces acrolein via the production of 3-AP. The background of autophagy activation observed in cell cultures mixed with bovine serum containing BSAO is similar to the background of autophagy activation observed in cells from the elderly and patients suffering from age-related diseases induced and promoted by chronic inflammation. Abbreviations: SMO, spermine oxidase; BSAO, bovine serum amine oxidase; 3-AP, 3-aminopropanal.

**Figure 6 biomedicines-11-02827-f006:**
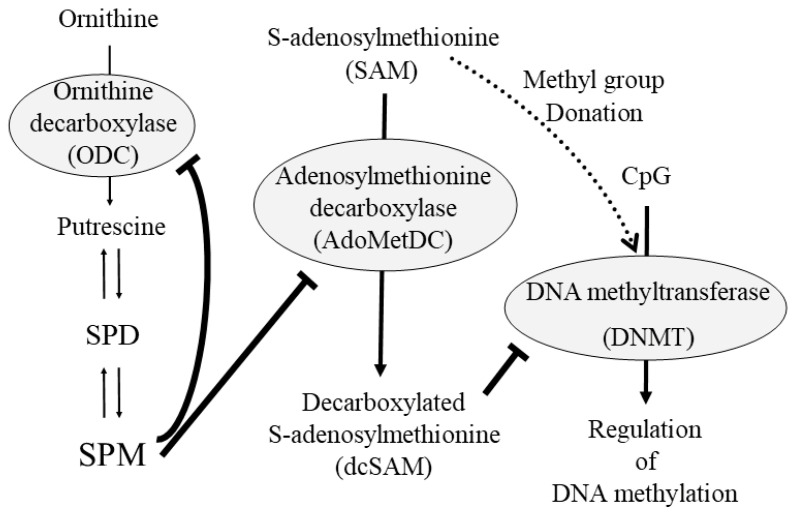
Close relationship between polyamine metabolism and gene methylation. SPM inhibits the polyamine synthesis enzymes, ODC and AdoMetDC. dcSAM inhibits DNMT activity. When dcSAM is reduced, the inhibitory effect of dcSAM on DNMT is reduced and DNMT is activated to regulate gene methylation. The solid arrows indicate the metabolic pathway, and the dashed black arrows indicate the transfer of methyl group from S-adenosylmethionine. The thick gray “T-bar” like lines indicate the inhibitory activity on target. Abbreviations: SPM, spermine; SPD, spermidine; ODC, ornithine decarboxylase; AdoMetDC, adenosylmethionine decarboxylase; dcSAM, decarboxylated S-adenosylmethionine; DNMT, DNA methyltransferase.

**Figure 7 biomedicines-11-02827-f007:**
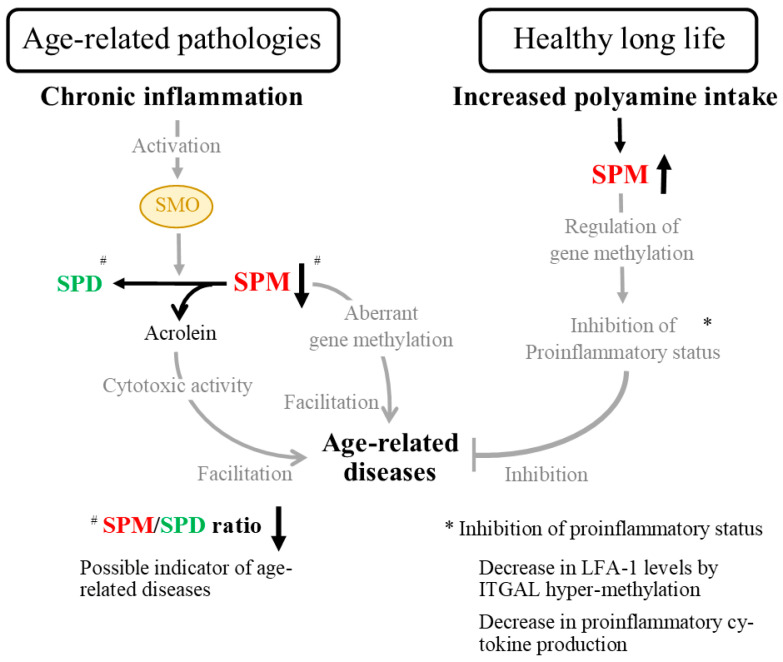
Polyamines and the background of healthy longevity and age-related diseases. For ease of understanding, spermine is listed in red, spermidine is listed in green, and SMO is circled in yellow. The gray lines and letters represent the action of each substance. Up and down thick linear arrows indicate concentration and ratio changes. (**left**): Chronic inflammation, thought to be a major factor underpinning age- and lifestyle-related diseases, activates a highly inducible enzyme, SMO. SMO breaks down SPM but not SPD. Thus, in an inflammatory environment, SPM decreases. Because SPM has stronger bioactivities than SPD, the result is a loss of methylation control, leading to aberrant methylation of the entire genome. In addition to the loss of inhibitory effects of SPM on inflammatory cytokine production, aberrant gene methylation is associated with increased demethylation of ITGAL, an LFA-1 protein gene. The demethylation of ITGAL increases LFA-1 protein levels, resulting in an increased proinflammatory status. In addition, a highly toxic substance, acrolein, produced by the enzymatic activity of SMO, exerts strong cytotoxic activity and consequently impairs cellular function. ^#^ The decrease in SPM and the increase in SPD due to the accelerated degradation of SPM results in a decrease in the SPM/SPD ratio. (**right**): On the other hand, blood SPM is increased via the continuous consumption of a high polyamine diet. SPM has a strong effect on the regulation of gene methylation, thereby inhibiting aberrant gene methylation. Controlling the methylation status of the entire genome is associated with increased methylation of the LFA-1 promoter region, resulting in decreased LFA-1 protein levels. In addition to the inhibition of proinflammatory cytokine production by SPM, decreased LFA-1 protein suppresses the age-related increase in proinflammatory status. Based on the accumulated evidence, the blood SPM/SPD ratio may be an indicator of healthy longevity and the onset and progression of age-related diseases. * The elements of “inhibition of proinflammatory status” are listed at the right bottom of the figure. I believe that the regulation of gene methylation and suppression of chronic inflammation are key to controlling age-related diseases. Abbreviations: SPM, spermine; SPD, spermidine; SMO, spermine oxidase; LFA-1, lymphocyte function-associated antigen 1.

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
