# Peer review of "Changes in Whole Blood Polyamine Levels and Their Background in Age-Related Diseases and Healthy Longevity"

_biomedicines, 2023, doi:10.3390/biomedicines11102827_

Round 1

Reviewer 1 Report

Recommendation: Major revision

Comments for the author

The review submitted by Kuniyasu Soda covers the field and contains interesting aspects. In many aspects, however, it remains vague (numerical data mostly are missing) and to some degree is outdated. This becomes apparent especially in Figures 1 and 2, which are old and oversimplified. In addition, the cytotoxicity of polyamine degradation products, if it really exists what is questionable, is overemphasized largely. In this case it appears as if Dr. Soda just refers to the literature without have read the deciding publications himself. Most important, the complete hypusine story is missing. Corrections and updating of the important flaws are urgently required.

Major concerns

1. Introduction

Page 1, line 42: …degradation products have been mistaken…

Nebulous formulation. Please explain in enough detail to let the reader understand.

2. Polyamines

Page 2, lines 58,59: … their intracellular concentrations…

The author should provide values for selected examples and corresponding referendes.

Page 2, line 66: … synthesis is initiated…

Such an oversimplification is not acceptable. Besides arginine and ornithine, the amino acids glycine, L-proline, L-glutamate are effective sources for polyamine (PA) biosynthesis (Du et  al, 2021; Lenis et al, 2017; Phang JM, 2012; Rieck et al, 2022). Furthermore, the agmatine pathway surpasses the ornithine step, directly providing putrescine (Rieck et al, 2022).

Page 3, Figure 1:

The scheme is oversimplified and important additional pathways are missing (see concern for line 66). Needs to be updated.

Page 3, line 101: …called the polyamine transporter…

The existence of a biological meaningful polyamine transporter is a matter of intense debate. More detail and references are necessary.

Page 4, Figure 2:

Again, the scheme is oversimplified and important additional pathways are missing (see Rieck et al, 2022). Needs to be updated.

Page 4, lines 117/118: …substances with potent cytotoxic activities.

With respect to adult, healthy animals, such ideas must be criticized. Most of the reports, which emphasize a high toxicity of PA degradation products, are based on experiments far away from physiological situations. Mostly they depend on the use of tissue culture, where peroxide or aldehydes often are added in grotesque amounts. Actually, experimental evidence for any damage produced by the degradation of PAs under physiological conditions is missing (Seiler, N, 2000).

Page 5, Figure 3, Legend line129: …dxhibit cytotoxic activity.

Unjustified formulation. See item above.

Page 4, lines 155 to 169:

Delete, see above.

Page 7, line 224: …aging and polyamine levels…

Polyamine levels where, in brain, in blood, in urine?

Page 8, line 254: …SPM/SPD ratio shows a tendency to decrease…

Weak formulation, give data!

Page 12, Summary Figure:

Figure very nebulous and partly desinformative.

E.g. column 1:

SPM down, how much? To what percent is gene methylation aberrant?

E.g. column 2:

Increased SPM, only marginal, which is not shown!

How strong is the inhibition of a proinflammatory status?

English language appears appropriate to me.

Author Response

Recommendation: Major revision

Comments for the author

The review submitted by Kuniyasu Soda covers the field and contains interesting aspects. In many aspects, however, it remains vague (numerical data mostly are missing) and to some degree is outdated. This becomes apparent especially in Figures 1 and 2, which are old and oversimplified. In addition, the cytotoxicity of polyamine degradation products, if it really exists what is questionable, is overemphasized largely. In this case it appears as if Dr. Soda just refers to the literature without have read the deciding publications himself. Most important, the complete hypusine story is missing. Corrections and updating of the important flaws are urgently required.

My Response

This review is not described to introduce polyamine metabolism. It is solely intended to enhance the reader's understanding of polyamine metabolism as it relates to healthy longevity and diseases that develop with aging. I understand that there are a variety of metabolic pathways, but I believe that covering these pathways will hinder the reader's understanding. You mentioned that Figures 1 and 2 are old. Yes, they are old, however the they are established. This review is not focused on the polyamine metabolism, but focused on the changes in polyamine concentrations in age-associated diseases and healthy longevity.

The production of toxic substances and their consequences on brain, kidney, stomach and heart in animals were reported by many researchers. Therefore, I (and many polyamine researchers) cannot agree with your comment “poyamine degradation products is questionable” and I don't think the term "the cytotoxicity of polyamine degradation products is overemphasized" is necessarily appropriate,

One basic scientific finding that can support the background of the relationship between changes in polyamine levels and age-related diseases is the regulation of gene methylation by polyamines. In my previous review, I described the action of polyamines in healthy longevity as involving regulation of gene methylation. In this review, I discuss the relationship between age-related diseases and methylation, which are inextricably linked to healthy longevity.

So, if the relationship between hypusin production and gene methylation is clear, and if polyamines regulate the methylation, we think it should be included in this review. We would very much appreciate information on this point.

Polyamines have a variety of physiological activities. Therefore, please understand that it is not possible to introduce each one of them in this review.

Major concerns

  1. Introduction

Page 1, line 42: …degradation products have been mistaken…

Nebulous formulation. Please explain in enough detail to let the reader understand.

My Response

There are papers done and published with incorrect experimental systems that it is impossible to point them all out. This is something that is often brought up at polyamine conferences. It is not a good practice to point out each and every one of them clearly. And, the degradation products are described in subsequent sentences and chapters.

  1. Polyamines

Page 2, lines 58,59: … their intracellular concentrations…

The author should provide values for selected examples and corresponding referendes.

My Response

It is so obvious that nothing is unknown among researchers. There are numerous references to it in other literature, and it would be undesirable to unnecessarily increase the number of references cited here. High intracellular concentrations are common knowledge among polyamine researchers.

Page 2, line 66: … synthesis is initiated…

Such an oversimplification is not acceptable. Besides arginine and ornithine, the amino acids glycine, L-proline, L-glutamate are effective sources for polyamine (PA) biosynthesis (Du et  al, 2021; Lenis et al, 2017; Phang JM, 2012; Rieck et al, 2022). Furthermore, the agmatine pathway surpasses the ornithine step, directly providing putrescine (Rieck et al, 2022).

My response

I understand your comment. However, again, the review is not focused on the polyamine metabolism. Therefore, better understanding of readers, relevant and important elements need to be picked up. Most importantly, polyamine researchers are conducting their experiments without knowing the enzymatic activity of bovine serum. It is the duty of polyamine researchers to disseminate the knowledge. I have revised to simplify the diagram because the readers may have similar opinion as you.

Page 3, Figure 1:

The scheme is oversimplified and important additional pathways are missing (see concern for line 66). Needs to be updated.

My Response

The additional pathways you tell me is a very new one. I do not think all polyamine investigators have a consensus. In fact, in the paper you indicated and papers cite in the paper, it seems to be expressed as presumably or may as for the role of its metabolites and enzymatic activity. On the other hand, they clearly described the production of toxic substances by polyamine degradation. Again, the review is not focused on the polyamine metabolism. I believe it would complicate and, therefore, be unwise to include details of polyamine metabolism.

Page 3, line 101: …called the polyamine transporter…

The existence of a biological meaningful polyamine transporter is a matter of intense debate. More detail and references are necessary.

My Response

O.K. I understand your comment. Again, this is not review of polyamine metabolism. Therefore, I deleted the word polyamine transporter, and related sentences. Line 97 in revised ms.

Page 4, Figure 2:

Again, the scheme is oversimplified and important additional pathways are missing (see Rieck et al, 2022). Needs to be updated.

My Response

Adding additional pathway blurs the point of this review and makes it difficult for the reader to understand the issue. In addition, in the paper and related referenced papers, it is often described that the relation of polyamine metabolites and enzymes described as “presumably or may be”.

This description means that it is not accepted many researchers and not established yet.

Page 4, lines 117/118: …substances with potent cytotoxic activities.

With respect to adult, healthy animals, such ideas must be criticized. Most of the reports, which emphasize a high toxicity of PA degradation products, are based on experiments far away from physiological situations. Mostly they depend on the use of tissue culture, where peroxide or aldehydes often are added in grotesque amounts. Actually, experimental evidence for any damage produced by the degradation of PAs under physiological conditions is missing (Seiler, N, 2000).

My response

I did not describe that the production of toxic substance in physiological condition. Toxic substances are produced in pathological conditions. Animal studies have clearly shown that polyamine degradation products are highly toxic in the brain, heart, stomach, and kidneys. Starting from 1990’s, many polyamine researchers have been published.

Page 5, Figure 3, Legend line129: …dxhibit cytotoxic activity.

Unjustified formulation. See item above.

My Response

I cannot understand your comment. The formulation is established one and consensus item. Many polyamine researchers know.

Page 4, lines 155 to 169:

Delete, see above.

My Response

I would like to start by saying that it is an incredible review comment to just ask us to remove it without any clear pointers. Erasing this statement is the same as erasing the scientific facts. The statements in this section are based on scientific facts and are not in any way erroneous. This is an important statement to help end the history of repeating the same mistakes for more than half a century. It is completely incomprehensible why this should be deleted without any explanation. The sentences are very important to aware young investigators who want to start polyamine research. It is not acceptable to be required to erase this factual statement based on the many previous reports.

This review article uses a considerably softer expressions in describing this section. However, the clear, direct, and strong descriptions in "Preprints" is already available online. Therefore, I hope to publish the paper with softer wording as soon as possible.

Page 7, line 224: …aging and polyamine levels…

Polyamine levels where, in brain, in blood, in urine?

My Response

I described in the ref. 6 paper. I do not believe it is necessary to increase the number of words by stating the same thing again.

Page 8, line 254: …SPM/SPD ratio shows a tendency to decrease…

Weak formulation, give data!

My Response

The figures can be easily verified by referring to the cited papers, and I don't think it is possible to publish similar tables and figures in this review because it would lead to double submissions.

Page 12, Summary Figure:

Figure very nebulous and partly desinformative.

E.g. column 1:

SPM down, how much? To what percent is gene methylation aberrant?

E.g. column 2:

Increased SPM, only marginal, which is not shown!

How strong is the inhibition of a proinflammatory status?

My Response

The name of the summary figure is changed to Figure 7

The upper and lower numbers are given in the cited paper. I am showing the overall trend and will not address the figures one by one as it would create a very busy figure and complicate the explanation.

The potent anti-inflammatory effects of spermine have already been confirmed in many papers, and we do not believe it necessary to reiterate the extent of these effects in this review.

Reviewer 2 Report

The submitted manuscript contains exactly the same data already proposed for another manuscript not accepted for publication by MDPI. I consider such action unprofessional.

Author Response

The submitted manuscript contains exactly the same data already proposed for another manuscript not accepted for publication by MDPI. I consider such action unprofessional.

My Response

I cannot agree with your comment “exactly the same”. This review article focuses on the relationship between polyamine concentrations and various age-related diseases that have been identified in recent years. The review focuses on just recently reported papers. The previous review focused on polyamines for healthy longevity, which is diametrically opposed to the current review. Admittedly, there are many similarities with my previous review, but I felt that a detailed explanation of polyamine metabolism was necessary to understand why polyamine concentrations are altered in age-related diseases. Polyamine researchers are concerned that some researchers are conducting studies without even understanding the presence of serum amine oxidase in bovine serum and are publishing very rudimentary and erroneous results.

The experimental design and incorrect interpretation itself resulting from such uninformedness is unprofessional. I have written this review in order to prevent such immature behavior from occurring. I would be grateful if you could help me understand. I'm sure if you read enough you will understand.

Reviewer 3 Report

This review succinctly describes polyamine catabolism and metabolism and how the levels of polyamines are related to age-related diseases.  In particular, I applaud the writer for describing how FBS (which can be incorrectly used in polyamine add back experiments) affects experimental outcomes in cases where exogenous polyamines are placed in the cell culture media.

Minor points:

Line 83:  Period at end of sentence.

Line 168:  SAPM should be changed to SPM.

Line 317:  I'm not sure what the author means by "in the figure".  Is that referring to the body of the animal?

Lines 436 and 439:  Call it Figure 7 and not Summary Figure.  Summary Figure can be added to the caption for clarity.

Overall, this is a well done review.  The figures are succinct and properly explained and there are few grammar issues.

Author Response

This review succinctly describes polyamine catabolism and metabolism and how the levels of polyamines are related to age-related diseases.  In particular, I applaud the writer for describing how FBS (which can be incorrectly used in polyamine add back experiments) affects experimental outcomes in cases where exogenous polyamines are placed in the cell culture media.

Minor points:

Line 83:  Period at end of sentence.

My Response

I added period at the end of sentence.

Line 168:  SAPM should be changed to SPM.

My Response

A in SAPM is deleted. Revised in red

Line 317:  I'm not sure what the author means by "in the figure".  Is that referring to the body of the animal?

My Response

I added “in the paper” after “in the figure”. Revised in red

Lines 436 and 439:  Call it Figure 7 and not Summary Figure.  Summary Figure can be added to the caption for clarity.

My Response

I changed the title of summary figure to figure 7.

Overall, this is a well done review.  The figures are succinct and properly explained and there are few grammar issues.

Response to Reviewer 3

Thank you very much for your evaluation. I assume you are a very trained polyamine researcher. I am relieved that you share my perception and understand the problems often encountered in polyamine research to date and appreciate the content of this review. All minor points noted have been corrected in red.

Round 2

Reviewer 1 Report

Comments for the author – responses to Dr. Soda

You mentioned that Figures 1 and 2 are old. Yes, they are old, however the they are established.

-   This is an extremely weak argument, which cannot be accepted.

The production of toxic substances and their consequences on brain, kidney, stomach and heart in animals were reported by many researchers.

-   Again, an extremely weak argument. The fact that several researchers have reported something cannot imply that their reports reflect the true biological situation. Most of the reports, which emphasize a high toxicity of PA degradation products, are based on experiments far away from physiological situations. Mostly they depend on the use of tissue culture, where peroxide or aldehydes often are added in grotesque amounts. Actually, experimental evidence for any damage produced by the degradation of PAs under physiological conditions is missing (Seiler, N. Neurochem. Res. 2000, 25, 471–490).

In animal models of injury, however, the situation may be different.

So, if the relationship between hypusin production and gene methylation is clear, and if polyamines regulate the methylation, we think it should be included in this review. We would very much appreciate information on this point.

-   Again, a rather funny argument. The relationship between hypusine and longevity is clear. To be aware of the available information is the job of the author.

Author Response

This is an extremely weak argument, which cannot be accepted.

My Response: Again, this review is not about polyamine metabolism in general; therefore, it is of utmost importance to provide the necessary and non-misleading information in this review. I believe that the details of polyamine metabolism should be described by another scientist. To describe everything about polyamine metabolism (including the possibilities) in this review would do nothing but confuse the reader. It is also important to point out the wrong experimental methods used by some uninformed researchers and to provide a good understanding of the basic background of polyamines so that no one will ever make such a foolish mistake again.

 Again, an extremely weak argument. The fact that several researchers have reported something cannot imply that their reports reflect the true biological situation. Most of the reports, which emphasize a high toxicity of PA degradation products, are based on experiments far away from physiological situations. Mostly they depend on the use of tissue culture, where peroxide or aldehydes often are added in grotesque amounts. Actually, experimental evidence for any damage produced by the degradation of PAs under physiological conditions is missing (Seiler, N. Neurochem. Res. 2000, 25, 471–490). In animal models of injury, however, the situation may be different.

My Response: What is far from physiological is that the changes caused by incubating polyamines in a culture medium mixed with bovine serum are physiological. Once again, many studies on animals and humans have pointed out that when pathological conditions occur, the toxic substances produced by polyamine degradation can cause various disorders. It saddens me as a polyamine researcher that these points are being made again. You must read lots of papers. Then, be a reviewer. Otherwise, it will only cause confusion and misunderstanding.

Again, a rather funny argument. The relationship between hypusine and longevity is clear. To be aware of the available information is the job of the author.

My Response: I don't think all scientists, myself included, are convinced of this. If you are so sure, please show me a paper that clearly demonstrates that hypsin activation extends the health and life span of “normal” humans and mammals. My review does not address the role of polyamines in lower primitive organisms or in vitro.

Reviewer 2 Report

Each of us researchers in the field of polyamines has his own ethical behavior. I have no doubts about the seriousness of my colleague Kuniyasu Soda. I consider my colleague to be a serious scientist, and I appreciate his willingness to sort out the information concerning the field of polyamines. For this reason, I am convinced that his work is useful for all of us.

Author Response

Thank you very much for your understanding of the purpose of the review.

Reviewer 3 Report

The authors have addressed my concerns.

Author Response

Thank you for conducting this thoughtful review.